# Global arthropod beta-diversity is spatially and temporally structured by latitude
Mathew Seymour [1] ✉, Tomas Roslin [2,3,4], Jeremy R. deWaard[5], Kate H. J. Perez[5], Michelle L. D'Souza[5], Sujeevan Ratnasingham[5], Muhammad Ashfaq [5], Valerie Levesque-Beaudin [5], Gergin A. Blagoev[5], Belén Bukowski [6], Peter Cale[7], Denise Crosbie[8], Thibaud Decaëns[9], Stephanie L. deWaard[5], Torbjørn Ekrem [10], Hosam O. El-Ansary[11], Fidèle Evouna Ondo[12], David Fraser[13], Matthias F. Geiger [14], Mehrdad Hajibabaei[5], Winnie Hallwachs[15], Priscila E. Hanisch[6,16], Axel Hausmann[17], Mark Heath[18], Ian D. Hogg[19,20], Daniel H. Janzen[15], Margaret Kinnaird[21], Joshua R. Kohn[22], Maxim Larrivée[23], David C. Lees[24], Virginia León-Règagnon[25], Michael Liddell [26], Darío A. Lijtmaer[6], Tatsiana Lipinskaya [27], Sean A. Locke[28], Ramya Manjunath[5], Dino J. Martins[29], Marlúcia B. Martins [30], Santosh Mazumdar[31], Jaclyn T. A. McKeown[5], Kristina Anderson-Teixeria[32], Scott E. Miller [32], Megan A. Milton [5], Renee Miskie[5], Jérôme Morinière[33], Marko Mutanen[34], Suresh Naik[5], Becky Nichols[35], Felipe A. Noguera[25], Vojtech Novotny [36,37], Lyubomir Penev[38], Mikko Pentinsaari[5], Jenna Quinn[39], Leah Ramsay[13], Regina Rochefort[40], Stefan Schmidt[17], M. Alex Smith[41], Crystal N. Sobel [5], Panu Somervuo[4], Jayme E. Sones[5], Hermann S. Staude [42], Brianne St. Jaques[5], Elisabeth Stur [10], Angela C. Telfer [5], Pablo L. Tubaro[6], Tim J. Wardlaw[43], Robyn Worcester[44], Zhaofu Yang[45,46], Monica R. Young [5,47], Tyler Zemlak[41], Evgeny V. Zakharov[5], Bradley Zlotnick[48], Otso Ovaskainen[4,49,50] & Paul D. N. Hebert[5]

Global biodiversity gradients are generally expected to reflect greater species replacement closer to the equator. However, empirical validation of global biodiversity gradients largely relies on vertebrates, plants, and other less diverse taxa. Here we assess the temporal and spatial dynamics of global arthropod biodiversity dynamics using a beta-diversity framework. Sampling includes 129 sampling sites whereby malaise traps are deployed to monitor temporal changes in arthropod communities. Overall, we encountered more than 150,000 unique barcode index numbers (BINs) (i.e. species proxies). We assess between site differences in community diversity using beta-diversity and the partitioned components of species replacement and richness difference. Global total beta-diversity (dissimilarity) increases with decreasing latitude, greater spatial distance and greater temporal distance. Species replacement and richness difference patterns vary across biogeographic regions. Our findings support long-standing, general expectations of global biodiversity patterns. However, we also show that the underlying processes driving patterns may be regionally linked.

Biodiversity is influenced by environmental, evolutionary, biotic, and stochastic processes, resulting in a global distribution of over 2 million described species[1], along with several million more undescribed species[2]. Global biodiversity is essential for life, in that it provides various environmental services, including energy and nutrient cycling, food security, and waste management, etc.[3,4]. As such, understanding what factors shape biodiversity across time and space, particularly at the global scale, is of interest to a wide range of researchers in ecology, evolutionary biology, conservation and invasive species management, agriculture, medical science and many others. It is generally accepted that biodiversity is expected to scale with latitude, increasing toward the tropics, a phenomenon referred to as the latitudinal diversity gradient (LDG)[5]. The underlying mechanism(s) for the LDG expectation are not definitive. Currently, there are over 30 hypotheses explaining the LDG, which are based on varying degrees of ecological,

evolutionary, and environmental complexity[6]. Recent efforts have also discovered exceptions to the LDG, which primarily indicate regional influences[7–10]. The many hypotheses advanced to explain global biodiversity patterns are also often difficult to test, particularly across taxonomic groups or at large spatial and temporal scales[11,12]. However, we can gain insights into biodiversity patterns using pairwise site assessment of total beta-diversity and its associated decomposition components.

The beta-diversity (i.e. β-diversity) framework provides a robust means to assess differences in biodiversity between communities, which can, in turn be used to determine spatio-temporal or environmental response[13]. Beta-diversity is the compositional difference (i.e. dissimilarity) between two communities[13]. This compositional dissimilarity between communities arises from two key processes: (i) species replacement (turnover), the change in community composition due to non-shared species, and (ii) richness difference (nestedness), the gain or loss of species between two communities. High species replacement can result from strong environmental forcing, competition, natural enemies, or historical disturbances[14]. Richness difference can be caused by species disappearing from a location (localized extinction), differing niche diversity, or other processes that result in the gain or loss of species[15,16]. Total beta-diversity, as per the beta-diversity framework from Podani et al.[17], can be separated into components of species replacement and richness difference, which sum to the total beta-diversity measure. Subsequently, the beta-diversity partitioning framework provides a means to investigate the potential links between global biodiversity patterns and the underlying processes associated with their formation across different species groups and ecological dimensions.

While several studies have investigated global biodiversity patterns, predominately using the LDG, they were constrained by three major limitations. First, they were mainly based on meta-analyses, as they combined data collected using different methodologies at different spatial and temporal resolutions, including seasonal variation[5,12]. While such data have high heuristic value, they are often affected by biases emerging from the varied sampling techniques underlying the individual data points[18]. Second, the few studies which have sampled communities using standardized methods to estimate differences in biological communities across broad latitudinal ranges have generally ignored the effects of temporal variability (i.e. seasonality) within or between sampled communities[19]. If differences in biological communities are only assessed across space, estimates of site-specific diversity will ignore the well-established importance of local spatiotemporal variation in describing patterns of biodiversity[20,21]. If site-specific diversity comparisons are made across different time points, the estimates of patterns of beta-diversity (i.e. diversity difference between sites) in space will alter patterns of beta-diversity in time, i.e. the scope for spatiotemporal interactions[22]. Third, prior studies have either examined less diverse taxa[5,22,23] or have generalized patterns emergent from local studies to the global scale[8,24]. Thus far, efforts to assess global patterns with standardized sampling methods have not been undertaken for taxonomic groups that comprise the bulk of global biodiversity.

With regards to the LDG, as the most prominent ecological assumption of global biodiversity distribution, species replacement is expected to increase at lower latitudes, reflecting greater habitat specialization and smaller ranges in more seasonally stable environments[13,25]. By contrast, richness difference is expected to increase with latitude, reflecting recent recolonization from a shared species pool following deglaciation[26]. Alternatively, species replacement may increase with latitude, which could reflect historical selection for species adapted for colder periods or to stronger changes in seasonality[27]. Different latitudinal patterns in richness difference, be it decreasing with latitude or unimodal, could indicate spatio-temporal disturbance patterns linked to regular or historical extinction events[28]. A lack of general patterns across multiple regions or continents may also indicate inconsistent patterns of global biodiversity, which may suggest that regional-specific processes, such as historical or environmental, predominate over expected environmental gradient filtering of community assembly. Regional ecosystems and their historical stability vary greatly across the planet, which has provided several key instances of unique adaptive radiations in response to specific environments[29]. Hence, comparisons between species replacement and richness difference can provide insights into the processes influencing global biodiversity patterns.

In this study, we adopted DNA-based methods to characterize beta-diversity for a highly diverse lineage of animals: terrestrial arthropods[30]. We comprehensively sampled 129 sites across the globe for an average of 22 sequential weeks each, encountering more than 150,000 different Barcode Index Numbers (BINs), which here serves as a species proxy[31]. We calculated and partitioned beta-diversity into its species replacement and richness difference components in both space and time to determine how global biodiversity patterns relate to latitude, distance, and time.

## Results

### The Global Malaise Trap Program data
The Global Malaise Trap Program (GMTP) was initiated in 2007 with the goal of observing global-scale spatiotemporal arthropod biodiversity dynamics (Figs. 1 and 2). Between 2010 and 2016, one or more Malaise traps were deployed at 129 sampling sites in 28 countries, with repeated weekly sampling ranging from 2 to 104 weeks (Fig. 2). The international collaboration of the Global Malaise Trap Program (Fig. 1; Supplementary Data 1) jointly produced the first set of global biodiversity data for terrestrial arthropods based on a uniform barcode sampling method. Details on GMTP standardized sampling protocols (e.g. trap type, sampling

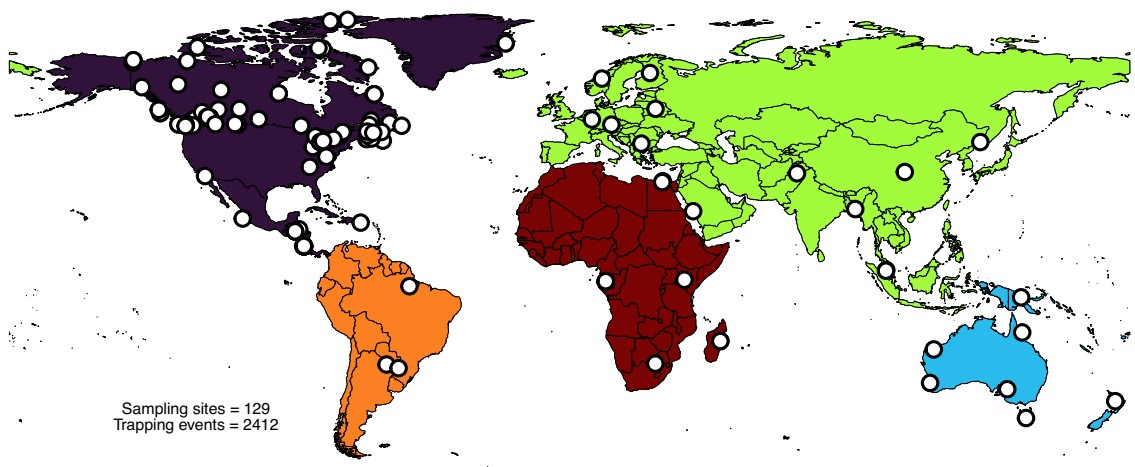

Sampling sites = 129
Trapping events = 2412

**Fig. 1 | Sampling sites and five biogeographical regions considered.** Regions are differentiated by color. White points indicate sampling locations.

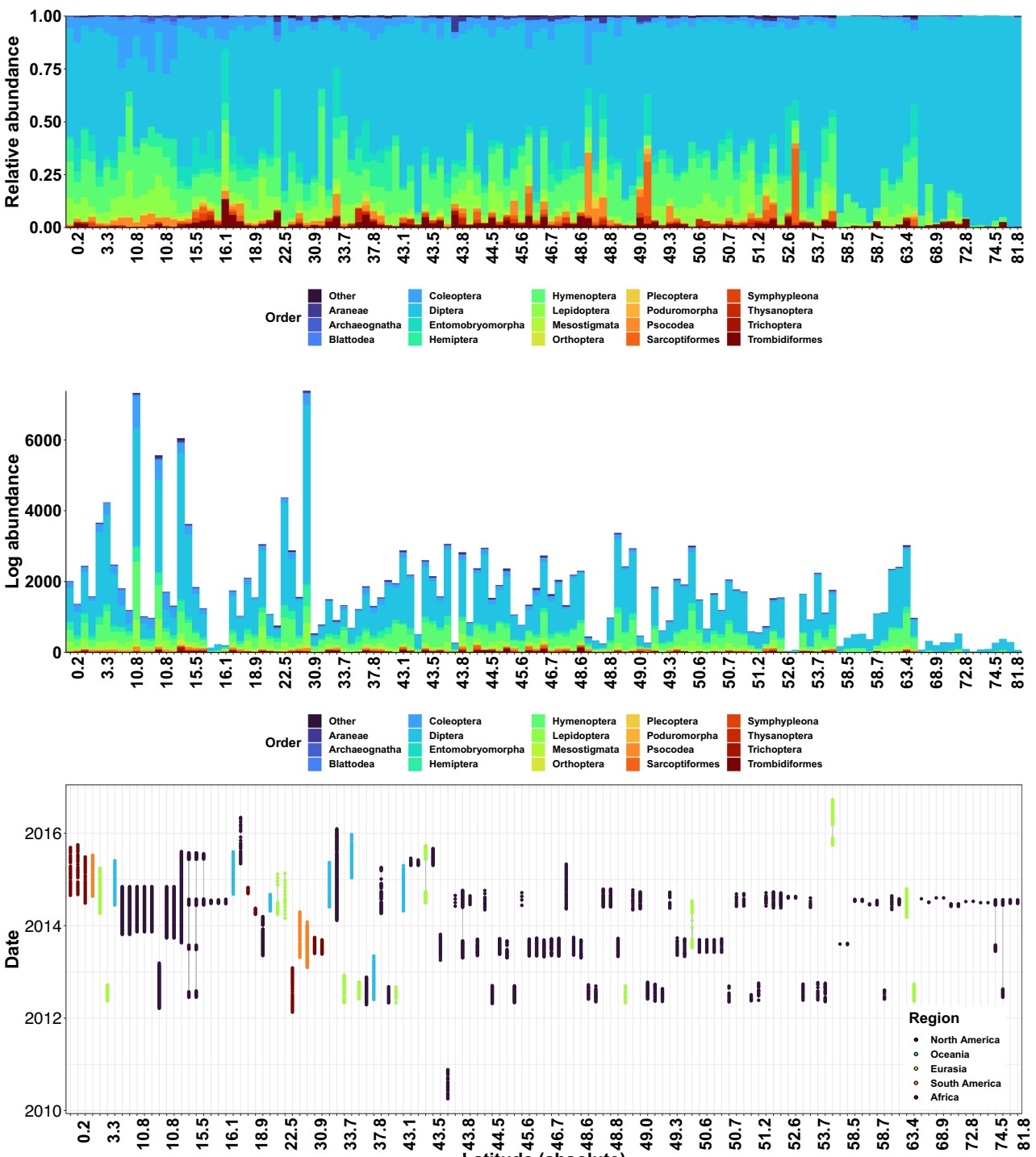

**Fig. 2 | Diversity of BINs (i.e. species proxies) across sampling sites.** Colors shown are unique terrestrial arthropod orders with each height corresponding to the relative abundance (top) or log abundance (middle) of unique BINs across the given site. The absolute latitude is provided on the x-axis with sites arranged from low to high. Bottom panel, each circle represents a unique sampling event, with colors corresponding to the region. Each vertical line shows a unique sampling location with latitude (absolute) indicated on the x-axis. Colors correspond to regional groups following the same color scheme as Fig. 1.

method, data curation) used for this study are outlined in detail in deWaard et al.[32].

Overall the GMTP recovered 155,185 unique BINs across five geographic regions (Fig. 1), representing regional variation in latitudinal, temporal, and spatial profiles (Fig. 2 and Supplementary Fig. 1). BIN diversity (Fig. 2) captured a large range of terrestrial diversity across 50 orders dominated by Diptera (51% of total BINs; 77,046 unique BINs), Hymenoptera (22% of total BINs; 33,265 unique BINs), Coleoptera (7% of total BINs; 12,550 unique BINs), Lepidoptera (7% of total BINs; 11,899 unique BINs), and Hemiptera (5% of total BINs; 7,783 unique BINs) (Fig. 2).

**Global and regional beta-diversity patterns**

Global beta-diversity (dissimilarity) decreased significantly with absolute latitude, which supports the general expectations of the LDG (Table 1; Supplementary Fig. 2). More generally, site comparisons between communities at higher latitudes were more similar to each other than site

**Table 1 | Statistical significance of patterns detected**

| Covariate | Region | Beta | Dir. | Richness difference | Dir. | Species replacement | Dir. |
|---|---|---|---|---|---|---|---|
| Latitude | Global | **<0.01** | − | 0.19 | + | 0.18 | − |
| Distance in space (ΔS) | Global | **<0.01** | − | 0.95 | + | 0.91 | − |
| Distance in time (ΔT) | Global | **<0.01** | − | 1.00 | − | 1.00 | + |
| ΔT × Latitude | Global | **<0.01** | + | **0.04** | + | **0.05** | − |
| ΔS × Latitude | Global | **<0.01** | + | **0.03** | − | **0.03** | + |
| Latitude | North America | **<0.01** | − | **0.02** | + | **0.01** | − |
| Distance in space (ΔS) | North America | **<0.01** | − | **0.01** | − | 0.45 | + |
| Distance in time (ΔT) | North America | 1.00 | − | **<0.01** | + | **<0.01** | − |
| ΔT × Latitude | North America | **<0.01** | + | 0.07 | − | **<0.01** | + |
| ΔS × Latitude | North America | **<0.01** | + | **0.01** | + | 0.63 | − |
| Latitude | South America | **<0.01** | + | **<0.01** | + | 0.17 | + |
| Distance in space (ΔS) | South America | 0.17 | − | 0.51 | − | 0.66 | + |
| Distance in time (ΔT) | South America | **<0.01** | + | 1.00 | + | **<0.01** | + |
| ΔT × Latitude | South America | 0.33 | − | 0.49 | + | 0.33 | − |
| ΔS × Latitude | South America | 0.85 | + | 0.15 | + | 0.19 | − |
| Latitude | Oceania | 0.71 | − | 0.90 | + | 0.81 | − |
| Distance in space (ΔS) | Oceania | 0.57 | + | 0.98 | + | 0.11 | − |
| Distance in time (ΔT) | Oceania | 1.00 | + | **<0.01** | − | 1.00 | + |
| ΔT × Latitude | Oceania | 0.33 | + | 0.37 | + | 0.45 | − |
| ΔS × Latitude | Oceania | 0.43 | + | 0.08 | − | 0.07 | + |
| Latitude | Eurasia | 0.13 | − | 0.29 | − | 0.37 | + |
| Distance in space (ΔS) | Eurasia | **<0.01** | + | 0.93 | − | **0.04** | + |
| Distance in time (ΔT) | Eurasia | 1.00 | + | **<0.01** | − | **<0.01** | + |
| ΔT × Latitude | Eurasia | 0.29 | + | 0.06 | + | **0.05** | − |
| ΔS × Latitude | Eurasia | 0.43 | + | 0.71 | + | 0.85 | − |
| Latitude | Africa | **0.03** | − | 0.83 | − | 0.40 | + |
| Distance in space (ΔS) | Africa | **<0.01** | + | **0.01** | + | **0.01** | − |
| Distance in time (ΔT) | Africa | 1.00 | + | 1.00 | + | **<0.01** | + |
| ΔT × Latitude | Africa | 0.52 | + | 0.60 | + | 0.88 | − |
| ΔS × Latitude | Africa | 0.91 | − | 0.54 | + | 0.22 | − |

To assess the importance of each candidate variable (listed on the left as Covariates), we used a series of permutation tests. We first calculated the log-likelihood ratio between the model where the explanatory variable being tested was included (the full model), and the model with the explanatory variable being tested was excluded (the reduced model). We then compared the observed log-likelihood ratio to its null distribution, which we computed by permuting the data $N = 1000$ times (see Methods for the exact permutation schemes implemented). This table shows the proportion of permutation outcomes for which the log-likelihood ratio of the model fitted to the actual data was lower than the log-likelihood ratio for the models fitted to the permuted data. Values at or below 0.05 are deemed significant and are indicated in bold and the direction of the response is indicated as Dir. +/−.

comparisons between communities closer to the Equator, with this observation also extending over longer temporal and spatial scales. Total beta-diversity spatio-temporal patterns were generally consistent in indicating latitudinal trends across biogeographic regions (Table 1; Supplementary Fig. 3). There were some deviations in patterns across the regions, particularly when assessing the partitioned components of species replacement and richness difference (Supplementary Fig. 2). Oceania did not have significant total beta-diversity or species replacement association with latitude, spatial distance or temporal distance, but richness difference was found to decrease significantly with increase temporal distance ($p < 0.01$) (Table 1). Eurasia did not have significant latitudinal relationships with total beta-diversity or its partitioned components but did show significant positive association between total beta-diversity dissimilarity and spatial distance ($p < 0.01$) along with significant positive associations between species replacement and spatial distance ($p = 0.04$), and temporal distance ($p < 0.01$), with a negative association with temporal distance × latitude ($p = 0.05$), as well as significant negative associations between richness difference and temporal distance ($p < 0.01$) (Table 1, Figs. 3 and 4). More generally, latitude, latitude × distance, or latitude × time were significantly associated with species replacement for two of five regions (North America

and Eurasia) and for one of five regions for richness difference (South America). In this regard latitude × distance interactions indicated that communities were increasingly more dissimilar with increasing distance at lower latitudes compared to higher latitudes. Similarly, latitude × time interactions indicated that communities were increasingly more dissimilar with greater difference in time at lower latitudes compared to higher latitudes. Spatial distance was significantly associated with species replacement for Africa ($p < 0.01$) and Eurasia ($p = 0.04$) with species replacement associated with latitude for North America ($p = 0.01$). Temporal distance or time × latitude was significantly associated with species replacement for four of five regions (except Oceania), with richness difference being significant for North America, Oceania, and Eurasia (Table 1; Figs. 3 and 4).

## Discussion
The dataset generated by the Global Malaise Trap Program offers a unique opportunity to assess the underpinnings of global latitudinal biodiversity patterns using a highly diverse and dominant group of terrestrial organisms. Our main finding is that total pairwise beta-diversity dissimilarity increases with decreasing latitude, increasing spatial distance, and increasing distance in time. We did not find strong indications of generalized

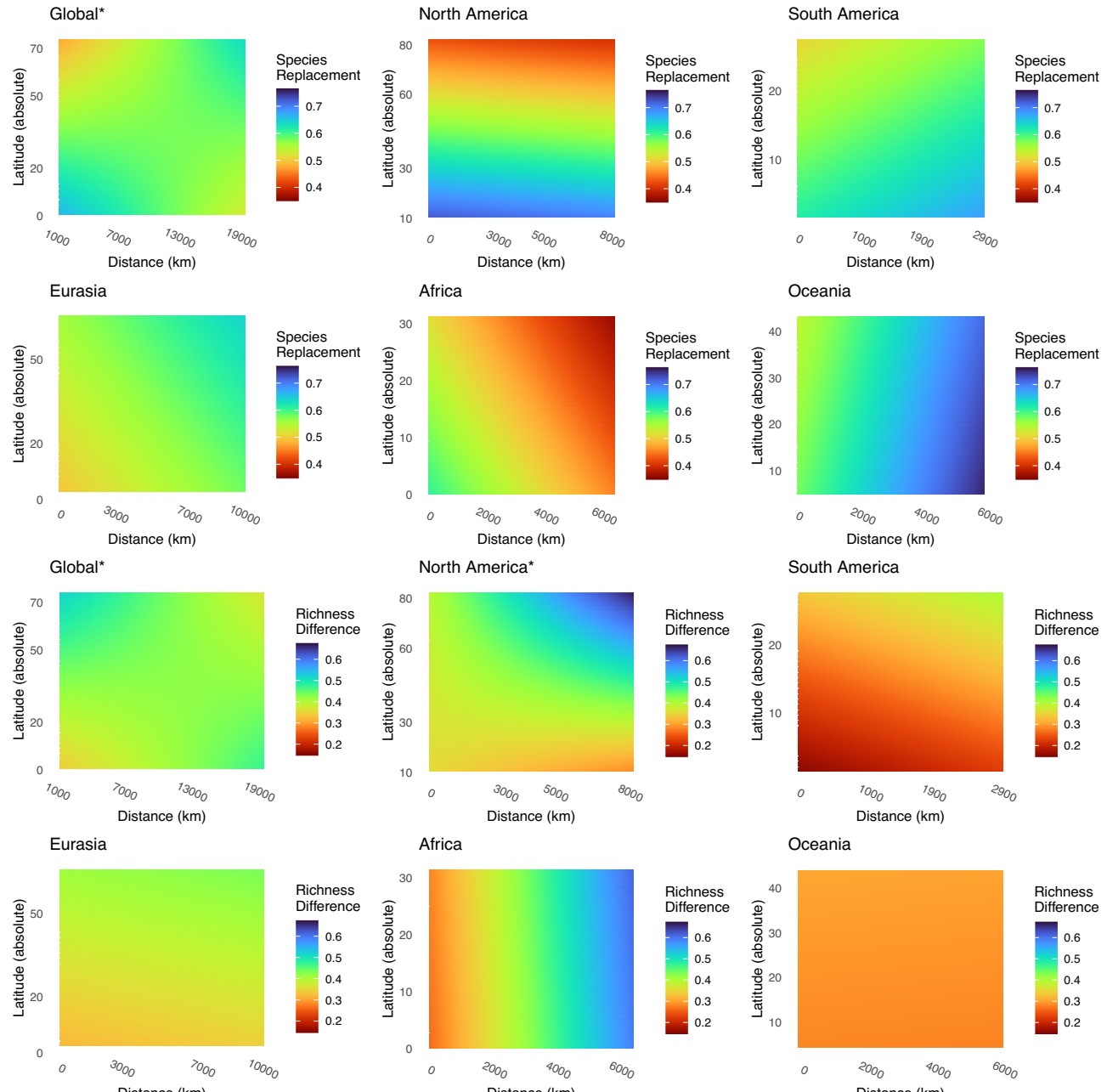

**Fig. 3 | Impacts of pairwise distance in space and latitude on community dissimilarity.** This figure shows the species replacement and richness difference components of beta-diversity, plotted from the fitted values of a linear model of the metric in question as a function of distance in space, distance in time (here set to zero), mean absolute latitude, and the interactions mean latitude × distance in space and mean latitude × distance in time (see Table 1 for statistical significances). In this figure, we explicitly test whether spatial patterns of community beta-diversity *in space* (in terms of overall beta-diversity, species replacement, or richness difference) varies detectably with latitude. Regions with a significant interaction between pairwise difference in latitude and pairwise distance are indicated by an asterisk. Note the differences in the scaling of axes among the individual graphs.

partitioned beta-diversity patterns (i.e. species replacement or richness difference) at the global scale. However, we did find partitioned beta-diversity patterns at the regional scale, which differed in the influence of latitude, spatial distance, and time indicating regional factors play a key role in overall beta diversity patterns (Figs. 3 and 4).

We found that Diptera contributed the most to insect diversity, accounting for 51% of the total BINs recovered. This finding challenges the commonly held belief that Coleoptera are the most biodiverse lineage of arthropods among regions[33]. In North America, Diptera dominated with 62% of the BINs, followed by Hymenoptera (13%), Lepidoptera (6%), and Coleoptera (5%). South America also showed a high occurrence of Diptera

(70%), along with Hymenoptera (7%), Hemiptera (7%), and Lepidoptera (5%). Africa exhibited a similar pattern, with Diptera (53%), Hymenoptera (18%), Hemiptera (9%), and Lepidoptera (8%) being the most prominent groups. Eurasia had a dominance of Diptera (60%), Hymenoptera (17%), and Hemiptera (6%), while Oceania displayed a collage dominated by Diptera (67%), Hymenoptera (10%), and Coleoptera (5%). These regional variations in the contribution of different taxa to insect diversity highlight the importance of considering local and global patterns in biodiversity. While Diptera emerged as the most diverse group in our study, further research and analysis will provide a more comprehensive understanding of the global patterns of arthropod biodiversity.

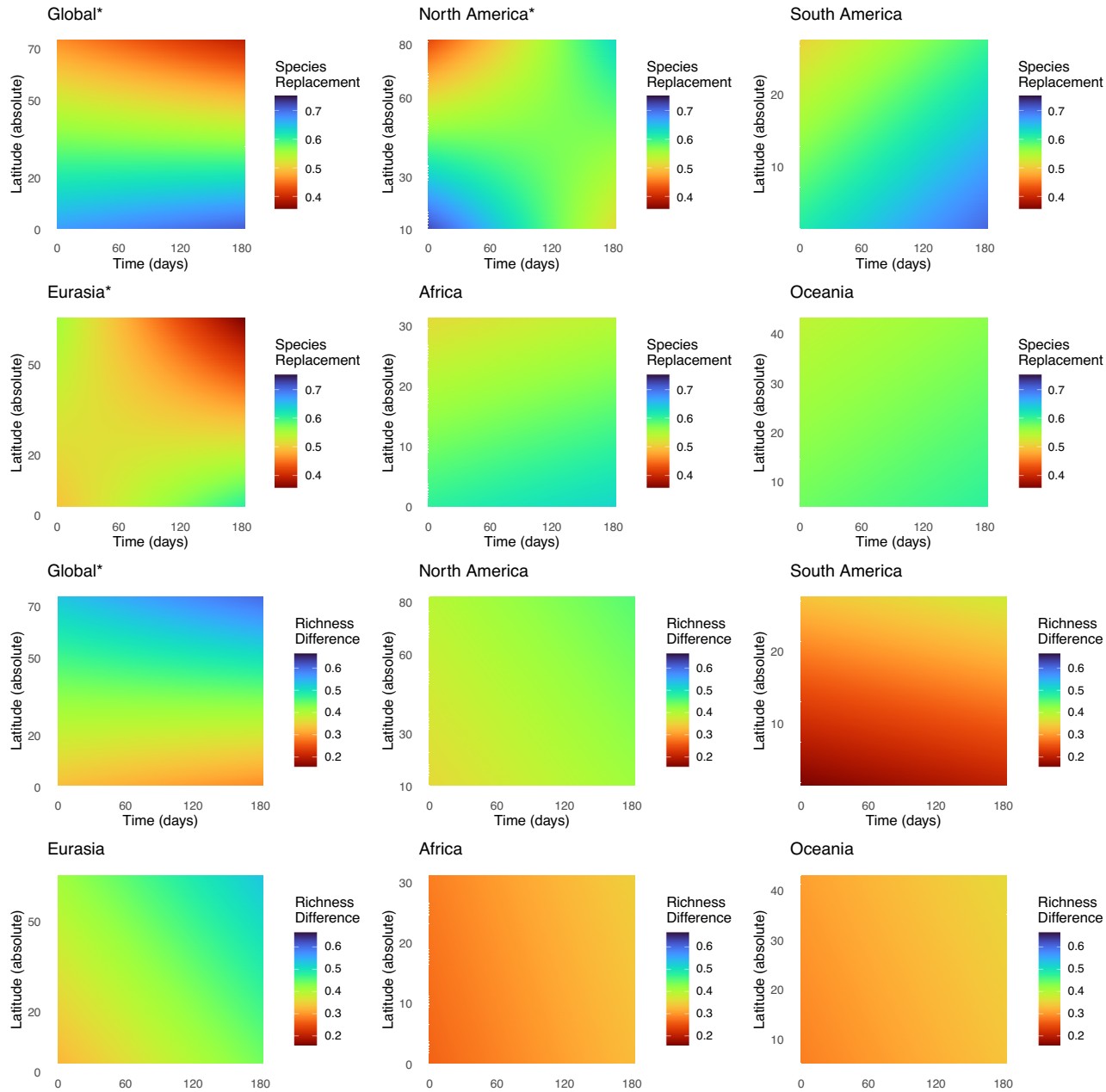

**Fig. 4 | Impacts of pairwise distance in time and latitude on community dissimilarity.** This figure shows the species replacement and richness difference components of beta-diversity, plotted from the fitted values of a linear model of the metric in question as a function of distance in space (here set to zero), distance in time, mean latitude, and interactions between mean latitude × distance in space and mean latitude × distance in time (see Table 1 for statistical significances). In this figure, we explicitly test whether temporal patterns of community beta-diversity *in time* (in terms of overall beta-diversity, species replacement, and richness difference) varies detectably with latitude.

To evaluate community dissimilarity in space and time, we partitioned overall beta-diversity into its components: richness difference and species replacement. This partitioning framework provides insights into what factors are driving differences in biological diversity between sites, and subsequently overall diversity gradients[13,34]. We expected richness difference to increase with increasing latitude, reflecting the recent recolonization of high latitude environments following deglaciation[26]. If species expanded their distributions from a joint source pool in a glacial refugium, communities established along the expansion front should form nested subsets of the source species pool. Consistent with this scenario, we generally found increased richness difference with increasing latitude (Table 1, Figs. 3 and 4). We also expected species replacement to decrease with increasing latitude as a direct effect of limiting factors or eco-evolutionary processes, leading to

smaller ranges and increased specialization in areas with less seasonal variability[25,35]. However, while we found clear support for latitudinal changes in overall beta-diversity, the underlying trends in species replacement and richness difference were inconsistent across regions, suggesting different, regional, ecological or evolutionary processes are influencing biodiversity patterns (Table 1, Figs. 3 and 4).

The appearance of generally consistent total beta-diversity patterns across geographic regions suggests alternatives to earlier observations pointing to regional rather than global factors influencing changes in community composition. However, the Oceania region seems to be an exception, as we found no association between changes in biodiversity community composition and spatial distance or latitude. This supports findings from Novotny et al.[24], which proposed that regional differences in

arthropod communities may actually be lower in Papua New Guinea than in temperate sites, indicating that beta-diversity is unlikely to explain the high diversity of tropical arthropod communities. However, their patterns and ours may suffer from lower site coverage than is needed to confirm the lack of latitudinal gradient, but it is interesting none the less.

The presence of inconsistent trends within a specific biogeographic region, such as Oceania, raises intriguing questions about the underlying processes driving local biodiversity patterns. Several factors may contribute to this disparity. Firstly, geological processes, such as the formation of mountains or islands, can create barriers that impede species dispersal and promote speciation[36]. These barriers can lead to the development of unique and isolated habitats within a region, resulting in distinct community composition. Secondly, ecological factors, such as competition, predation, and resource availability, can vary across different localities within a region, which directly influence local and regional biodiversity patterns[37]. Additionally, climatic factors, including temperature, precipitation, and seasonality, can vary within a region, creating diverse regional microhabitats leading to patterns of localized environmental filtration[38], or longer term, region-specific, diversification of species through evolutionary processes[39]. Finally, area-effect processes, such as habitat fragmentation or island size, can also influence species richness and biodiversity patterns within a region[40].

Past studies have largely focused on temperate sites, drawing some concern that tropical sites may be underestimated in assessing LDG dynamics. The GMTP data includes 30 tropical sampling sites (23% of the sampled sites) over 986 trapping events (40.1% of the total trapping effort), providing a wider assessment compared to previous assessment. We did find partitioned components of beta-diversity differed across regions, however, which would suggest that although the general global spatial trends in beta-diversity are consistently observed, the underlying environmental or biotic drivers of species replacement and richness difference may be regional rather than global[11]. The more frequent association with spatial distance versus latitude, particularly outside North America might also support recent indications that the strength of the LDG may be greater in the western hemisphere[12]. It is also worth noting that the regional trends observed in our study may be reflective of regional differences in paleological climatic stability, particularly for South America and Africa, influencing local and regional species evolution and biological diversity[41]. Additionally, local and regional topography is known to play a crucial role in shaping alpha- and beta-diversity patterns, whereby unique local habitats can lead to unique species evolution and specialization, promoting elevated alpha-diversity within a given region[42]. Topographic heterogeneity can influence climatic conditions by altering temperature and precipitation gradients, further altering alpha-diversity patterns[43,44]. As such regional topography has a direct effect on beta-diversity by influencing habitat connectivity and subsequently species dispersal. Whereas the LDG provides a parsimonious assessment of general biodiversity trends, considering the regional barriers (e.g. mountains, deserts, oceans) and connectors (e.g. rivers or ocean currents) with regards to time and space are essential for robust assessments of biodiversity patterns.

While our analytical approach accounted for the presence of small sample sizes for some regions, particularly South America and Africa, additional sampling of these regions is needed to validate the regional latitudinal and spatial trends observed here. Given the importance of regional aspects to biodiversity trends found in this study, future sampling efforts should seek to assess biodiversity across regional variation in habitat, landuse, elevation and seasonality to build upon our findings, as well as others.

In stark contrast to previous studies, especially in assessing global scale patterns, our analyses also considered temporal effects on latitudinal beta-diversity dynamics. Temporal species replacement or richness difference were significant for all five regions, including the global scale (Table 1). Temporal ecological dynamics are important for understanding seasonal shifts in habitat and home ranges which can influence spatial biodiversity patterns[21,45]. While pronounced temporal changes in environmental

conditions (e.g. seasonality) in the temperate zone have been well documented, the same may also be true for the tropics. Temporal species replacement was noted for several regions, indicating seasonal shifts in community composition likely due to competition and seasonal effects of environmental forcing[46,47]. Here we note variation in rainfall, radiation, leaf flush, etc. has been proposed to generate strong seasonality in the activity of arthropods[48,49]. Temporal richness replacement, which predominantly was co-associated with a significant effect of latitude likely attributed to species loss during key seasonal shifts, which may be more prominent at higher latitudes where seasonal shifts in environmental conditions are greater[47]. The observed temporal patterns here attest to finer partitioning of community composition that should be accounted for in determining the mechanistic associations with larger spatial/latitudinal biodiversity patterns.

The prevalence of high beta-diversity values between sites, particularly the high number (88%) of pairwise global sites that were completely dissimilar (i.e. shared no species) highlights the extraordinary diversity of terrestrial arthropods. Communities become more similar at the regional level with 63% dissimilar sites for North America, 69% for Oceania, 50% for Eurasia, 51% for Africa, and 34% for South America. While most arthropods can disperse by flight, both individual home ranges and species distributions are commonly restricted[50]. Range sizes have been proposed to shrink towards the tropics, following Rapoport's rule. Nonetheless, the evidence for this assertion is very limited as it has traditionally been derived from studies predominately conducted in the Northern Hemisphere[22], but some recent support for Southern Hemisphere trends are available[51,52]. Our observation of greater species replacement may reflect greater niche partitioning and specialization allowed by higher productivity or stemming from greater levels of speciation[5].

While previous assessments of beta-diversity have largely involved regional assessments[21,47], which were then used to fuel meta-analyses[5,12], this study represents a true global assessment of temporal-spatial dynamics of the most diverse lineage of terrestrial animals. The consistency in general global patterns which were decomposed at the regional scale enables a mechanistic assessment of the planetary biodiversity patterns. This synthesis was only made possible by our coupling of a standardized sampling method with DNA-based taxonomic assignments[53]. Importantly, convincing analyses of beta diversity require an efficient means for rigorously establishing the incidence of species shared across sites in massive sampling programs. Such methods are finally available for our use on a planetary scale.

## Methods

Arthropods were captured using a standard Townes-style Malaise trap deployed at each sampling location (hereafter site), with arthropods harvested from each trap weekly (hereafter trapping event). Traps were set up primarily in designated conservation areas (108 out of 129 sites). Habitat type was predominantly forest ($N = 70$), but also included were grassland ($N = 15$), tundra ($N = 14$), wetland ($N = 9$), urban ($N = 5$), and mixed habitat ($N = 16$) (Supplementary Fig. 4). Arthropod specimens captured from each trapping event were sorted, photographed, and processed individually. Analysis began with each specimen identified morphologically to a taxonomic order and registered on the Barcode of Life Data Systems (BOLD). DNA from each specimen was then extracted and used to amplify and Sanger sequence the standard cytochrome c oxidase I (COI) barcode region[53]. The resulting COI sequence data was uploaded to the BOLD database, linking each specimen's morphological identification to its COI barcode sequence. For each trapping event, all specimens were sequenced, except when a particular morphospecies was represented by more than 50 individuals, in which case a subset of the individuals were sequenced to confirm that the specimens did indeed represent a single unique BIN[32]. The final GMTP dataset includes 1.2 million barcode records and 155,185 unique barcode index numbers (BINs)[31]. Prior studies have established a strong correspondence between BINs and species identification in insect groups with well-established taxonomy, thereby justifying the recognition of BINs as species proxies[31].

Temporal differences (i.e. distance in time) between each pair of sampling events was calculated using circular statistics by first determining the Julian day of the two sampling events and taking two measures (1) the absolute difference between the two Julian days divided by 0.986 (0.986 degrees = 1 day) and (2) 360 minus the absolute difference between the two Julian days divided by 0.986. The minimum value between measure (1) and (2) was then used as the distance in time between the two sampling events. Here we refer to difference in time which includes changes in seasons since the study, and sampling period for individual sampling sites, spans multiple seasons (Fig. 2) and since seasonality differs drastically between different global locations. Distance between each pair of sampling locations (i.e. distance in space) was calculated as the geographic distance between site pairs using the function distHaversine in the R package geosphere[54]. Mean absolute latitude was calculated between each pair of sites along the LDG. To understand how this metric behaves, consider a site pair in which both members are at the Equator. In this case, their mean absolute latitude is 0° - which also applies to two samples from the same trap at the Equator. For a trap pair at the North Pole, the mean absolute latitude is 90°N; for a trap pair with its members on the North vs South Pole, it will be 90°, and for a trap pair of which one member sits on the North Pole and the other at the Equator, mean absolute latitude will be 45°N.

## Statistics and reproducibility

Community data were converted to presence absence data for calculations and analyses of diversity. Beta-diversity and its components were calculated as Jaccard dissimilarity using the Podani family of indices, which is a "true" beta-diversity estimate that is unaffected by the species pool (i.e., gamma-diversity)[15]. Total beta-diversity (here Jaccard dissimilarity) and the associated components of species replacement and richness difference were calculated for each site pair using the function beta.div.comp in the R-package *adespatial*[13]. We do note that there are alternative beta-diversity partitioning methods[13,15]. The Podani family was utilized here as it does not overestimate diversity differences and provides a "true" diversity estimate that is unaffected by the total species pool[13,15], but see also alternative true-diversity based partition approaches[55,56]. Utilizing a "true-diversity" allows for independent measures of alpha, beta, and gamma diversity. Whereas alpha-diversity reflects within site variation, beta-diversity may either reflect between site variation independently or dependently (i.e. scaling with alpha-diversity) depending on the measure used[57]. Using an independent (i.e. "true") measure of beta-diversity becomes more important when comparing beta-diversity measures. Measures of beta-diversity dependent on alpha-diversity may compromise interpreting results that actually reflect within-site instead of between site observations[58].

Pairwise values of total beta-diversity, species replacement, and richness difference were calculated for all trapping event pairs by taking the lower triangle values from the associated distance matrix. As there were 2412 trapping events in total, $N = 2{,}907{,}666$ pairs of trapping events were included in our analyses.

Using linear regression, we modeled each pairwise beta-diversity component as a separate, univariate function of distance in space, distance in time, mean latitude, and the interactions mean latitude × distance in space and mean latitude × distance in time. Here, the two interaction terms are of key interest in explicitly testing whether the rate of beta-diversity, species replacement or richness difference in space or time, respectively, varies detectably with latitude.

Our data are not fully balanced as the number of data points per site, and hence pairs of sites, varies. In the analyses, we wished to give each site, and pair of sites, an equal weight in the analyses. If $n_{s_1 s_2}$ is the number of pairs of trapping events for which one trapping event belongs to site $s_1$ and the other trapping event belongs to site $s_2$, in an unweighted regression this pair of sites would achieve the total weight of $n_{s_1 s_2}$, and thus sites with more data would contribute disproportionally to our analyses. To account for the unbalanced sampling effort in our models, we applied a weighted linear regression, where the weight for each data point was set to $1/n_{s_1 s_2}$, so that the total weight was equal among all pairs of sites.

We note that data points are not independent of each other, because each data point in the linear model involves a pair of samples that are correlated in time and space. For this reason, we did not perform significance tests based on output from the linear model but instead employed the following permutation approach to determine significance for each of the explanatory variables in our models.

Given the unequal temporal sampling and spatial sampling design across the multiple GMTP project datasets we used a series of permutation test to assess the significance of each explanatory variable[59]. For each permutation test we assessed the significance of each explanatory variable individually, including distance in space, distance in time, mean latitude, and the interactions mean latitude × distance in space and mean latitude × distance in time, by doing the following. We first calculated the log-likelihood ratio between the model where the focal explanatory variable was included (the full model), and the model where the focal explanatory variable was excluded (the reduced model). We compared the observed log-likelihood ratio to its null distribution which we computed by permuting the data $N = 1000$ times, with the permutation scheme detailed below for each specific test. In general, if the log-likelihood ratio for the full vs. reduced model fitted to the actual data was greater than the log-likelihood ratio for the full vs. reduced model fitted to the permuted data for at least 95% of the permutation outcomes, the explanatory variable was deemed significant[59].

When testing for the interaction between mean absolute latitude and distance in space, we permuted the sampling sites, keeping all trapping events that belonged to the same original site in the same group. When testing for the interaction between mean absolute latitude and distance in time, we first permuted the sampling sites as described above and permuted the sampling dates within each group of trapping events. When testing for the main (non-interactive) effects of the explanatory variables, we reduced the full model to exclude the respective interaction associated with the explanatory variable being tested. When testing for the main effect of distance in space or for the main effect of mean absolute latitude, we permuted the sampling sites. When testing for the main effect of distance in time, we permuted the dates within sampling locations.

## Data availability

All data associated with the manuscript, including source data for the analysis and figure generation, are provided in Supplementary Data 1, Supplementary Data 2 and Supplementary Data 3.

## Code availability

R scripts for data processing, analyses and figure generation can be found at https://github.com/MatSeymour/MyWebsite/tree/GMTP_R-code[60].

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

## Acknowledgements

M.S. was funded by the Hong Kong University Grants Committee, Early Career Scheme (grant no 27108123). T.R. and O.O. were funded by Academy of Finland (grant no. 322266 and 309581 respectively), Jane and Aatos Erkko Foundation and the European Research Council (ERC) under the European Union's Horizon 2020 research and innovation programme (grant agreement No 856506; ERC-synergy project LIFEPLAN). B.B., P.D.N.H., P.T. and D.L. acknowledge support from the International Development Research Centre of Canada, the Richard Lounsbery Foundation, and the Consejo Nacional de Investigaciones Científicas y Técnicas (CONICET) of Argentina. H.O.E. extends his deep appreciation to Researchers Supporting Project (RSP2024R118), King Saud University, Riyadh, Saudi Arabia. M.F.G. and J.M. with the GBOL project were generously supported by a grant from the German Federal Ministry of Education and Research (FKZ 01LI1101 and 01LI1501). M.Li and T.J.W. acknowledges the support of the Daintree Discovery Centre staff and the Australian Government's Terrestrial Ecosystem Research Network (TERN; www.tern.org.au) in maintaining the field infrastructure at Cow Bay and Warra. V.N. was supported by the Czech Grant Agency (grant no. 19-28126X). B.Z. acknowledges ResMed Incorporated and the San Diego Barcode of Life. D.H.J. and W.H. were funded by the University of Pennsylvania, USA, the Guanacaste Dry Forest Conservation Fund, the government of Costa Rica, under permit R-008-2018 OT-CONAGEBIO, JICA, Japan International Collaboration Agency and the Wege Foundation of Grand Rapids, Michigan. O.O. was also funded by the Research Council of Norway through its Centres of Excellence Funding Scheme (223257). P.D.N.H. and colleagues were supported by the Government of Canada through awards from the New Frontiers in Research Fund and the Canada First Research Excellence Fund as well as by Genome Canada and Ontario Genomics (Large Scale Applied Research Program, International Consortium Initiative). The analytical and informatics platforms required to support data acquisition and analysis at the Centre for Biodiversity Genomics were supported by awards from the Canada Foundation for Innovation and the Ontario Ministry of Research and Innovation. We kindly thank David Wagner and five anonymous reviewers for their assessment and feedback on the manuscript.

## Author contributions

M.S., T.R., J.R.d.W., K.H.J.P., M.L.D., O.O. & P.D.N.H. Designed the study, T.R., J.R.d.W., K.H.J.P., M.L.D., S.R., M.A., V.L.B., G.A.B., B.B., P.C., D.C., T.D., S.L.d.W., T.E., H.O.E.A., F.E.O., D.F., M.F.G., M.H., W.H., P.E.H., A.H., M.He., I.D.H., D.H.J., M.K., J.R.K., M.L., D.C.L., V.L.R., M.Li, D.A.L., T.L., S.A.L., R.M., D.J.M., M.B.M., S.M., J.T.A.M., K.A.T., S.E.M., M.A.M., R.Mi., J.M., M.M., S.N., B.N., F.A.N., V.N., L.P., M.P., J.Q., L.R., R.R., S.S., M.A.S., C.N.S., P.S., J.E.S., H.S.S., B.S.J., E.S., A.C.T., P.L.T., T.J.W., R.W., Z.Y., M.R.Y., T.Z., E.V.Z. & B.Z. carried out data collection and preprocessing, M.S., T.R. & O.O. analyzed the data, M.S., T.R., J.R.d.W., K.H.J.P., M.L.D., O.O. & P.D.N.H. drafted the manuscript. All authors contributed to the revisions.

## Competing interests

The authors declare no competing interests.

## Additional information

[1]School of Biological Sciences, The University of Hong Kong, Pokfulam Road, Hong Kong SAR, China. [2]Department of Ecology, Swedish University of Agricultural Sciences (SLU), Ulls väg 18B, Uppsala 75651, Sweden. [3]Department of Agricultural Sciences, Faculty of Agriculture and Forestry, University of Helsinki, PO Box 27 Helsinki, Finland. [4]Organismal and Evolutionary Biology Research Programme, Faculty of Biological and Environmental Sciences, University of Helsinki, P.O. Box 65 Helsinki 00014, Finland. [5]Centre for Biodiversity Genomics, University of Guelph, Guelph, ON, Canada. [6]División Ornitología, Museo Argentino de Ciencias Naturales "Bernardino Rivadavia" (MACN-CONICET), Buenos Aires, Argentina. [7]Australian Landscape Trust, Renmark, SA SA5341, Australia. [8]The Wetlands Centre, Cockburn, WA WA6163, Australia. [9]CEFE, Univ Montpellier, CNRS, EPHE, IRD, Montpellier, France. [10]Department of Natural History, NTNU University Museum, Norwegian University of Science and Technology, Trondheim NO-7491, Norway. [11]Plant Production Department, College of Food & Agriculture Sciences, King Saud University, Riyadh 11451, Saudi Arabia. [12]Agence Nationale des Parcs Nationaux, Departement de la Recherche Scientifique, Libreville, Gabon. [13]BC Conservation Data Centre, Ministry of Environment, Box 9338Station Prov Govt, Victoria, BC V8W 9M1, Canada. [14]Leibniz Institute for the Analysis of Biodiversity Change, Museum Koenig Bonn,

Adenauerallee 160, 53113 Bonn, Germany. [15]Department of Biology, University of Pennsylvania, Philadelphia, PA 19104, USA. [16]Department of Animal Ecology and Tropical Biology, Biocenter - University of Würzburg, Am Hubland, 97074 Würzburg, Germany. [17]SNSB—Zoologische Staatssammlung München, Munich, Germany. [18]Private collector, Perth, Australia. [19]Canadian High Arctic Research Station, Polar Knowledge Canada, Cambridge Bay, NU, Canada. [20]School of Science, University of Waikato, Hamilton, New Zealand. [21]Worldwide Fund for Nature- International, Nairobi, Kenya. [22]Section of Ecology, Behavior and Evolution, School of Biological Sciences, University of California San Diego, 9500 Gilman Drive, La Jolla, CA 92093-0116, USA. [23]Insectarium, Montréal Space for Life, Montréal, QC, Canada. [24]Department of Science, Natural History Museum, South Kensington, London, United Kingdom. [25]Estación de Biología Chamela, Instituto de Biología, Universidad Nacional Autónoma de México, A. P. 21, C.P, 48980 San Patricio, Jalisco, Mexico. [26]Centre for Tropical, Environmental, and Sustainability Sciences, James Cook University, Cairns, Queensland, Australia. [27]Laboratory of Hydrobiology, Scientific and Practical Center for Bioresources, National Academy of Sciences of Belarus, Minsk, Belarus. [28]Departamento de Biología, University of Puerto Rico at Mayagüez, Mayagüez 00680, Puerto Rico. [29]Mpala Research Centre and Department of Ecology & Evolutionary Biology, Princeton University, Princeton, NJ, USA. [30]Laboratório de Ecologia de Invertebrados, Coordenação de Zoologia, Museu Paraense Emilio Goeldi, Avenida Perimetral 1901, Terra Firma, CEP, 66077 530 Belém, Pará, Brazil. [31]Department of Zoology, University of Chittagong, 4331 Chittagong, Bangladesh. [32]National Museum of Natural History, Smithsonian Institution, Washington, WA, USA. [33]AIM - Advanced Identification Methods GmbH, Leipzig, Germany. [34]Ecology and Genetics Research Unit, University of Oulu, PO Box 3000, 90014 Oulu, Finland. [35]US National Park Service, 1316 Cherokee Orchard Road, Great Smoky Mountains National Park, Gatlinburg, TN, USA. [36]Biology Centre, Czech Academy of Sciences, Institute of Entomology, Ceske Budejovice, Czech Republic. [37]Faculty of Science, University of South Bohemia, Ceske Budejovice, Czech Republic. [38]Institute of Biodiversity and Ecosystem Research, Bulgarian Academy of Sciences, 2 Gagarin Street, 1113 Sofia, Bulgaria. [39]Rare Charitable Research Reserve, Cambridge, ON, Canada. [40]North Cascades National Park Service Complex, 810 State Route 20, Sedro-Woolley, WA 98284, USA. [41]Department of Integrative Biology, University of Guelph, Guelph, ON, Canada. [42]Lepsoc Africa, Magaliesburg, South Africa. [43]ARC Centre for Forest Values, University of Tasmania, Hobart, TAS, Australia. [44]Stanley Park Ecology Society, P.O. Box 5167 Vancouver, BC V6B 4B2, Canada. [45]Key Laboratory of Plant Protection Resources and Pest Management, Ministry of Education, Northwest A&F University, Yangling 712100 Shaanxi, China. [46]Entomological Museum, College of Plant Protection, Northwest A&F University, Yangling 712100 Shaanxi, China. [47]Canadian National Collection of Insects, Arachnids and Nematodes, Agriculture and Agri-Food Canada, Ottawa, ON, Canada. [48]San Diego Barcode of Life, San Diego, CA 92130, USA. [49]Department of Biological and Environmental Science, University of Jyväskylä, P.O. Box 35 (Survontie 9C), FI-40014 Jyväskylä, Finland. [50]Department of Biology, Centre for Biodiversity Dynamics, Norwegian University of Science and Technology, Trondheim N-7491, Norway. ✉e-mail: matsey@hku.hk

