## [Peer Review File · Communications Biology]

Reviewers' comments:

Reviewer #1 (Remarks to the Author):

This study uses DNA metabarcoding data to assess the beta-diversity of arthropods. The dataset includes information from 129 sites and identified over 150,000 barcodes that can be considered species proxies. This data is used to assess the latitudinal biodiversity gradient hypothesis for terrestrial arthropod beta-diversity. Beta-diversity is assessed in relation to latitude, space and time and is partitioned into its two major components: species replacement and richness difference (i.e. turnover and nestedness).

Overall, global patterns seem to support the latitudinal biodiversity gradient, however, there are differing responses for the regions assessed. I found the results challenging to follow and I think some of the supplementary figures show the main results more clearly than those in the main text. The main results are more clearly expressed in the discussion.

The shared code and data is not in a format that that would enable anyone to recreate the analyses or reuse the data for further analyses.

Comments:

Abstract:

The abstract is generally clear, although a little more detail on the sampling and analysis methods used would be helpful to set the scene.

Line 130: "drawing on samples across" – I think it would be helpful to expand slightly here – samples of what, and collected how?

Keywords:

Add some if you have space to aid searches – arthropods, beta-diversity, turnover

Introduction:

Line 147-148: I am suggesting some rewording here just to make things read better: Biodiversity is influenced by environmental, evolutionary, and stochastic factors as well as biotic interactions, resulting in ...

Line 148: 2 million is the rough number of named species but the estimated total number of species is much higher. Perhaps consider some clarification here or use an estimate of total species on the planet rather than just described species.

Line 150: suggested rewording, replace "in the form of various environmental services" with "in that it provides various environmental services".

Line 158: replace "evolution" with "evolutionary".

Line 196-196: patterns in spatiotemporal or environmental responses? Is this what you mean here?

Line 173: I would move the "(i.e. nestedness) on line 176 to this line so that it is consistent across definitions.

Line 188: it is not clear what the "e.g. seasonality is linked to in this first point. Do you mean that data were collected across different seasons but this was not accounted for? Please clarify.

Line 200: i.e. there is scope for spatiotemporal interactions

Line 227: again the reason for the e.g. seasonality is not clear here. It is not clear whether beta-diversity over time is being assessed or whether the affect of data collection during different seasons is being assessed.

Methods:

Figure 2: I think this might be very challenging to see in print. Please increase the size of the font. I think the latitude labels could be made less frequent so that there is more space for larger font.

Figure 2 legend: It looks like some editing is needed to clarify the legend. For example: "top and middle panels: (each vertical bar)." What does this mean?

Figure 2 legend: replace "sampling unique event" with "unique sampling event". Also where are the circles?

Figure legend, line 700: This sentence doesn't make sense. There are no horizontal lines, latitude is on the x-axis.

Line 244: CIO barcode region, can this acronym be explained?

Line 260 – 262: I'm not sure I follow this line. How are you incorporating seasonality?

Line 289-290: should this read "because of the potential dependence of alpha and beta diversity". It is not clear what this part between commas is for.

Results:

Line 348: Here beta-diversity (similarity) is described as similarity increasing with latitude, however in the abstract, it is described as "beta-diversity (dissimilarity) increasing with decreasing latitude". Can there be a consistent representation of the main results please, it was a little confusing!

Figure S3: There is a spelling mistake in beta-diversity in the figure legend. It also says richness replacement rather than richness difference. I think this figure would be better in the main text as it clearly shows the results and is more visually appealing than table 1. Although it is challenging to see what is significant or not here without relating to the table as there are no confidence intervals, could these be added? Just something to consider.

Figure S4: Please fix overlapping values on the x axis. Spelling mistakes in figure legend – respective and illustrative.

Line 362: Should the p-value = 0.04 as you state that it is significant in the sentence?

Code availability – it is not clear from looking at the github page whether all the code for the analysis is present. As far as I can tell only the permutation test code is there. It is not clear how figures were created. The readme is empty. There are no comments in the code at all. Neither of the two scripts are explained in any way. I suggest this is improved. Ideally an external user could run the analyses and recreate the results from the paper.

Data availability – a csv of the data is supplied as well as a PDF (from which data cannot be read). It is not clear what each of these files are. The csv only contains 4 columns: project code (which is not translated anywhere, it is not present in table S1), BIN (with no associated meta-data), Order and family. In this state I don't see how any one else could use this data. Meta-data should be provided. The code refers to a number of datasets Counttables , MetaDataSite, and MetaDataTrapEvent. It is not clear what these refer to.

Reviewer #2 (Remarks to the Author):

The paper 'Global arthropod beta-diversity is spatially and temporally structured by latitude' describes the outcome of an ambitious approach to sample arthropods at the global scale using a consistent sampling protocol and DNA barcoding. The analyses focus on beta diversity (total and decomposed into species replacement and differences in species richness) and the effects of latitude, spatial distance, temporal difference and their interactions.

The dataset is exceptional. Yet, it comes with some inevitable limitations. Some regions particularly South America had only few sampling sites which limits the ability to analyze the region separately. With such small n, the results probably become sensitive to the inclusion of single sites and I think this should be addressed in the discussion.

I am completely missing a description of the habitat type in which the traps were set up. This should be included in the Methods. Actually, it could be also very interesting to evaluate the effect of habitat differences in the analyses.

I think it would be nice to have a similar figure as Fig 3 and 4 for overall beta diversity at least for the global scale because it helps to understand how patterns of replacement and richness differences add up in total

List of minor comments:

L 126: maybe clarify to a broad audience what you mean by species replacement

L 132: unclear: do you mean differences in diversity or community composition?

L147: The drivers of biodiversity are again mentioned in line 152. I would drop this part of the sentence in line 147 and address it more in line 152f

L281: ...as it does...

L348: in the methods in in line 360 you say dissimilarity?

L364: If the word count allows, I think it would be nicer for the reader to always say in a few words what the interactions mean: eg dissimilarity increased more strongly with spatial distance at higher than lower latitude

Fig1: It is not relevant for the analyses, just the figure, but it looks strange that the island of New Guinea belongs to two zones. Maybe better use a biogeographical delineation rather than one based on country borders?

Reviewer #3 (Remarks to the Author):

The amount of data that sits behind this study is Herculean. The taxonomic, spatial, and temporal breadth and depth have no previous equal as far as I am aware: 129 sites across the globe with more than 155,000 species-level taxa enumerated. The subject matter: global biodiversity patterns will be of interest to a broad range of biological disciplines. The basic findings of the study reinforce those reported for plant and vertebrate taxa, and thus add credibility to the study's results. That patterns often broke down or disappeared at the regional level was also a interesting message (see below). Given the volume of data and focus of the study, the number of taxa, and number of ecological guilds represented, I expect the work will be highly cited and have a long shelf-life.

It is also worth noting that the team of authors includes many highly respected scientists that have collectively published over a thousand peer-reviewed papers, many of which have appeared in high-profile journals. Their collective knowledge about biodiversity, entomology, Malaise traps, barcoding, BINS, and statistical analyses speaks to the soundness of the study.

I am not sufficiently knowledgeable of the statistics used to evaluate the methods. That Seymour and others on the team have published other works employing biodiversity metrics, bolsters the team's choice of methods/approach. Much of the data and code have been made available but a bit more detail would be appreciated. Are COI sequences available for all the taxa?; what trap samples or specimen vouchers remain?; were DNA extracts saved?

The figures are fine.

As an insect biosystematist, I was also intrigued to see the numbers of BINS for the various arthropod taxa. The study provides an important view of what the tree of life actually looks like relative to what one sees in textbooks, i.e., that god had an inordinate fondness for midges and gnats—not beetles as is commonly heralded. And by extension, implies that Diptera probably have a more fundamental role in global food webs and nutrient cycling than heretofore understood.

Most of my comments have been added to the narrative which I am attaching. Below I comment on a few matters.

I would not call the regional differences in patterns stochastic, as the biotas of the continents/realms/regions have been shaped by natural selection and biogeographic laws over millennia. While seemingly idiosyncratic, all are surely explainable given more data about the region and attributes of where the Malaise trap samples were run. I would call the within and across regional differences syncretistic not stochastic. The former embraces the idea that each region has a different history and set of forcings.

One factor that I believe is important in determining both alpha diversity and beta diversity that was not mentioned was the paleoclimatic/historical stability of an ecosystem or region. I just returned from Mozambique. I was stunned by the number of flightless beetles—which no doubt = elevated beta diversity. There are almost no flightless beetles in Connecticut, a state that was recently glaciated. This can only happen where a site has been largely stable through antiquity. I recall hearing the same is true for Hemiptera in South Africa, i.e., unparalleled degree of flightlessness there. Stability promotes both alpha and beta diversity. The species diversity and tiny range sizes of plants across the fynbos is a third example. Some of the exceptional differences in regional patterns could relate to antiquity + stability within and across these areas.

Another major determinant of local/regional alpha and beta diversity is topography, and such could have a major influence of within and between regional patterns. Brazil and Columbia may have the richest biodiversity in the world because they have biotically distinct mountain ranges. Africa has only modest topography by comparison. Topographical differences surely underlie part of the “stochasticity” in the results. Might species/area (island biogeography) account for the non-conforming results from Oceania?

I bring these up as a way of reinforcing my point that the results from different regions likely have sound biogeographic/ecological explanations and that they are not so much stochastic as syncretistic in nature.

A bit more discussion about what is driving the non-conforming regional patterns would be welcome. Overall the manuscript is Spartan, with just one page of results and < three pages of discussion. Perhaps a bit more discussion about the patterns, drivers, exceptions, and taxonomic implications would be welcome.

Might some self-assessment of the limitations of the study would be helpful to future workers. Sampling was sparse in South America and Africa. It doesn't appear traps were run in the more diverse (mountainous) regions of either continent. What results were most perplexing and could be better evaluated with more data? I would be interested in getting the author's views of what would be needed to get why some of the regional patterns did not conform to expected patterns.

Minor matters: alpha and beta diversity are not hyphenated. Hyphen use across the manuscript needs to be considered by someone skilled in the Queen's English (grin)—that would not be me. I noticed several novel and incorrect usages across the introduction.

In same vein, when to use between, among, and across at various points in the narrative could be tuned up. Any comparisons involving more than two entities should be using among (or across) rather than between.

Southern Hemisphere should be capitalized; especially if Equator is capitalized. Both are recommended by Chicago Style Manual.

Lines 147-149: The estimate of over 2 million is ridiculously low and out of step with modern estimates. At the very least it should be adding the descriptor "metazoans." Or it should say more than 1.5 million described species. This is an arthropod paper and thus is better grounded to Stork (2018). The number is well north of 5.5 million.

Stork, N. E. (2018). How many species of insects and other terrestrial arthropods are there on Earth? *Annual Review of Entomology*, 63, 31-45.

García-Robledo, C. E. K. Kuprewicz, C. S. Baer, E. Clifton, G. G. Hernández, and D. L. Wagner. 2020. The Erwin Equation of Biodiversity: From little steps to quantum leaps in the discovery of tropical diversity. *Biotropica*. 52: 590-597.

Overall, this is a terrific effort of wide interest and can go forward with little modification.

David Wagner

Reviewers' comments:

Reviewer #1 (Remarks to the Author):

This study uses DNA metabarcoding data to assess the beta-diversity of arthropods. The dataset includes information from 129 sites and identified over 150,000 barcodes that can be considered species proxies. This data is used to assess the latitudinal biodiversity gradient hypothesis for terrestrial arthropod beta-diversity. Beta-diversity is assessed in relation to latitude, space and time and is partitioned into its two major components: species replacement and richness difference (i.e. turnover and nestedness).

Overall, global patterns seem to support the latitudinal biodiversity gradient, however, there are differing responses for the regions assessed. I found the results challenging to follow and I think some of the supplementary figures show the main results more clearly than those in the main text. The main results are more clearly expressed in the discussion.

The shared code and data is not in a format that that would enable anyone to recreate the analyses or reuse the data for further analyses.

RESPONSE: Thank you. We have addressed the specific comments below. We have also added additional annotation for the code and data to help with useability for interested readers.

Comments:

Abstract:

The abstract is generally clear, although a little more detail on the sampling and analysis methods used would be helpful to set the scene.

RESPONSE: We have included additional information on the sampling and analysis methods. We have formatted the abstract in accordance with *Communication Biology's* manuscript standards.

Line 130: "drawing on samples across" – I think it would be helpful to expand slightly here – samples of what, and collected how?

RESPONSE: We have included that malaise traps were used to collect arthropod communities. "Sampling included 129 sampling sites whereby malaise traps were deployed to monitor temporal changes in arthropod communities."

Keywords:

Add some if you have space to aid searches – arthropods, beta-diversity, turnover

RESPONSE: We have included turnover. Arthropods and beta-diversity are part of the title.

Introduction:

Line 147-148: I am suggesting some rewording here just to make things read better: Biodiversity is influenced by environmental, evolutionary, and stochastic factors as well as biotic interactions, resulting in ...

RESPONSE: Thank you, we have edited the sentence as suggested.

Line 148: 2 million is the rough number of named species but the estimated total number of species is much higher. Perhaps consider some clarification here or use an estimate of total species on the planet rather than just described species.

RESPONSE: We have edited this to indicate that 2 million is the number of known species. Additionally, also in line with reviewer #3, we indicate that there are several million more undescribed species (lines 143-145).

Line 150: suggested rewording, replace “in the form of various environmental services” with “in that it provides various environmental services”.

RESPONSE: We have edited the sentence to read, “...in that it provides various environmental services.”

Line 158: replace “evolution” with “evolutionary”.

RESPONSE: Edited as suggested.

Line 196-196: patterns in spatiotemporal or environmental responses? Is this what you mean here?

RESPONSE: Yes, here we are referring to spatiotemporal dynamics. We have edited the sentence to emphasize this. It now reads, “If differences in biological communities are only assessed across space, estimates of site-specific diversity ignore the well-established importance of local spatiotemporal variation in describing patterns of biodiversity.” (lines 186-189)

Line 173: I would move the “(i.e. nestedness) on line 176 to this line so that it is consistent across definitions.

RESPONSE: The “(i.e. nestedness)” has been moved two lines up.

Line 188: it is not clear what the “e.g. seasonality is linked to in this first point. Do you mean that data were collected across different seasons but this was not accounted for? Please clarify.

RESPONSE: We have edited the text to read, “temporal resolution, including seasonal variation”

Line 200: i.e. there is scope for spatiotemporal interactions

RESPONSE: We believe the current wording is correct, i.e. “the scope for spatiotemporal interactions.”

Line 227: again the reason for the e.g. seasonality is not clear here. It is not clear whether beta-diversity over time is being assessed or whether the affect of data collection during different seasons is being assessed.

RESPONSE: We have deleted “(e.g. seasonality)” to avoid confusion.

Methods:

Figure 2: I think this might be very challenging to see in print. Please increase the size of the font. I think the latitude labels could be made less frequent so that there is more space for larger font.

RESPONSE: Figure edited with increased font size and reduced number of x axis labels, as suggested

Figure 2 legend: It looks like some editing is needed to clarify the legend. For example: “top and middle panels: (each vertical bar).” What does this mean?

RESPONSE: The text has been deleted to avoid confusion, as it is redundant with the following words in the legend.

Figure 2 legend: replace “sampling unique event” with “unique sampling event”. Also where are the circles?

RESPONSE: Edited as requested. Circles have been changed to points to better illustrate.

Figure legend, line 700: This sentence doesn’t make sense. There are no horizontal lines, latitude is on the x-axis.

RESPONSE: Edited the legend text to read x-axis and vertical line instead.

Line 244: CIO barcode region, can this acronym be explained?

RESPONSE: The full name for the common acronym has been provided as cytochrome c oxidase I

Line 260 – 262: I’m not sure I follow this line. How are you incorporating seasonality?

RESPONSE: The traps were set so that sites sampled across seasons. Previous reviewers were confused as to why we did not mention seasonality, so we included the term. Here we again clarify by editing the text to read, “Here we refer to difference in time which includes changes in seasons since the study, and sampling period for individual sampling sites, spans multiple seasons.”

Line 289-290: should this read “because of the potential dependence of alpha and beta diversity”. It is not clear what this part between commas is for.

RESPONSE: This sentence has been broken into two sentences to help with clarity. It now reads, “Using an independent (i.e. “true”) measure of beta-diversity becomes more important when comparing beta-diversity measures. Measures of beta-diversity dependent on alpha diversity may compromise interpreting results that reflect within-site instead of between site observations.”

Results:

Line 348: Here beta-diversity (similarity) is described as similarity increasing with latitude, however in the abstract, it is described as “beta-diversity (dissimilarity) increasing with decreasing latitude”. Can there be a consistent representation of the main results please, it was a little confusing!

RESPONSE: We have edited the text throughout to indicate dissimilarity when referring to beta-diversity and removed the term “similarity.”

Figure S3: There is a spelling mistake in beta-diversity in the figure legend. It also says richness replacement rather than richness difference. I think this figure would be better in the main text as it clearly shows the results and is more visually appealing than table 1. Although it is challenging to see what is significant or not here without relating to the table as there are no confidence intervals, could these be added? Just something to consider.

RESPONSE: Figure S3 only provides the standardized model coefficient, which is an indication of the change in the response variable and not an indication of whether the variables or interactions are significant. As such, the table provided in the main manuscript gives a more thorough description of the results, including the permutation results, whereas the figure is better utilized as a supplement so as not to unintentionally mislead readers. The standard errors are included, they are just very small. We have added additional legend text to Figure S3 and edited the figure to improve aesthetics and readability as suggested.

Figure S4: Please fix overlapping values on the x axis. Spelling mistakes in figure legend – respective and illustrative.

RESPONSE: The supplementary figure has been edited as suggested.

Line 362: Should the p-value = 0.04 as you state that it is significant in the sentence?

RESPONSE: Yes, p-values less than 0.05 are considered significant as the sentence states.

Code availability – it is not clear from looking at the github page whether all the code for the analysis is present. As far as I can tell only the permutation test code is there. It is not clear how figures were created. The readme is empty. There are no comments in the code at all. Neither of the two scripts are explained in any way. I suggest this is improved. Ideally an external user could run the analyses and recreate the results from the paper.

RESPONSE: We have provided additional annotation to the existing data analysis files. There was no readme file associated with the code, though we now provide one indicating that the code is associated with the analysis provided in the manuscript. We have also now provided the scripts used to generate the figures.

Data availability – a csv of the data is supplied as well as a PDF (from which data cannot be read). It is not clear what each of these files are. The csv only contains 4 columns: project code (which is not translated anywhere, it is not present in table S1), BIN (with no associated meta-data), Order and family. In this state I don't see how any one else could use this data. Meta-data should be provided. The code refers to a number of datasets Counttables , MetaDataSite, and MetaDataTrapEvent. It is not clear what these refer to.

RESPONSE: The pdf is provided via Communication Biology's platform, where we initially submitted the supplement in csv format. The Trap Event and ProjectCode columns from the two files are the same set of codes to allow matching across the files. We have now simplified this to a single descriptor. TrapEvent, as well as included SiteNumber to further link both datasets. We have provided more details to make the code easier to use via the Readme file. The countable is generated from the now separated first script "00.GenerateCounttables.R," which takes the supplementary file to generate.

Reviewer #2 (Remarks to the Author):

The paper 'Global arthropod beta-diversity is spatially and temporally structured by latitude' describes the outcome of an ambitious approach to sample arthropods at the global scale using a consistent sampling protocol and DNA barcoding. The analyses focus on beta diversity (total and decomposed into species replacement and differences in species richness) and the effects of latitude, spatial distance, temporal difference and their interactions.

The dataset is exceptional. Yet, it comes with some inevitable limitations. Some regions particularly South America had only few sampling sites which limits the ability to analyze the region separately. With such small n, the results probably become sensitive to the inclusion of single sites and I think this should be addressed in the discussion.

RESPONSE: We agree and have explicitly addressed the unbalanced sampling design in the methods and results (lines 424-455). Specifically, we utilized a permutation test and weighted model that considered the variation in sampling size when generating the model test (lines 424-455). We have provided additional text in the discussion (263-268) to further clarify that additional sampling of these regions would improve the trends found here.

I am completely missing a description of the habitat type in which the traps were set up. This should be included in the Methods. Actually, it could be also very interesting to evaluate the effect of habitat differences in the analyses.

RESPONSE: Trap locations were mostly set up in conservation areas (N=108 out of 129 sites) and forested (N=70 of 129) to minimize habitat/landuse effects on sampling. This is now provided in the methods (lines 349-352), and we provide the habitat type as part of the supplementary file. We also provide a supplementary figure showing the breakdown of the habitat type in relation to species richness collected across the regions (Figure S6).

I think it would be nice to have a similar figure as Fig 3 and 4 for overall beta diversity at least for the global scale because it helps to understand how patterns of replacement and richness differences add up in total

RESPONSE: This is what Figure S2 and S4 are showing more directly. We have now also included the total-beta version of 3 and 4 as a supplement figure S5.

List of minor comments:

L 126: maybe clarify to a broad audience what you mean by species replacement

RESPONSE: The abstract length is limited to 150 words. Clarification of the beta-diversity components are already given in full detail on lines 162-180.

L 132: unclear: do you mean differences in diversity or community composition?

RESPONSE: Diversity, as the text is written.

L147: The drivers of biodiversity are again mentioned in line 152. I would drop this part of the sentence in line 147 and address it more in line 152f

RESPONSE: The text has been edited.

L281: ...as it does...

RESPONSE: Edited

L348: in the methods in in line 360 you say dissimilarity?

RESPONSE: Changed to dissimilar

L364: If the word count allows, I think it would be nicer for the reader to always say in a few words what the interactions mean: eg dissimilarity increased more strongly with spatial distance at higher than lower latitude

RESPONSE: We have added additional text to clarify the interactions in more general terms (line 244-250)

Fig1: It is not relevant for the analyses, just the figure, but it looks strange that the island of New Guinea belongs to two zones. Maybe better use a biogeographical delineation rather than one based on country borders?

RESPONSE: We have edited the figure to include the island of New Guinea colored as a whole

Reviewer #3 (Remarks to the Author):

The amount of data that sits behind this study is Herculean. The taxonomic, spatial, and temporal breadth and depth have no previous equal as far as I am aware: 129 sites across the globe with more than 155,000 species-level taxa enumerated. The subject matter: global biodiversity patterns will be of interest to a broad range of biological disciplines. The basic findings of the study reinforce those reported for plant and vertebrate taxa, and thus add credibility to the study's results. That patterns often broke down or disappeared at the regional level was also an interesting message (see below). Given the volume of data and focus of the study, the number of taxa, and number of ecological guilds represented, I expect the work will be highly cited and have a long shelf-life.

It is also worth noting that the team of authors includes many highly respected scientists that have collectively published over a thousand peer-reviewed papers, many of which have appeared in high-profile journals. Their collective knowledge about biodiversity, entomology, Malaise traps, barcoding, BINs, and statistical analyses speaks to the soundness of the study.

I am not sufficiently knowledgeable of the statistics used to evaluate the methods. That Seymour and others on the team have published other works employing biodiversity metrics, bolsters the team's choice of methods/approach. Much of the data and code have been made available but a bit more detail would be appreciated. Are COI sequences available for all the taxa?; what trap samples or specimen vouchers remain?; were DNA extracts saved?

RESPONSE: Thank you for your assessment. We address each of your points fully below. COI sequence data and voucher photos were not used in this study directly but are available via the BOLD system database.

The figures are fine.

RESPONSE: Thank you

As an insect biosystematist, I was also intrigued to see the numbers of BINS for the various arthropod taxa. The study provides an important view of what the tree of life actually looks like relative to what one sees in textbooks, i.e., that god had an inordinate fondness for midges and gnats—not beetles as is commonly heralded. And by extension, implies that Diptera probably have a more fundamental role in global food webs and nutrient cycling than heretofore understood.

Most of my comments have been added to the narrative which I am attaching. Below I comment on a few matters.

RESPONSE: Thank you for your assessment here.

I would not call the regional differences in patterns stochastic, as the biotas of the continents/realms/regions have been shaped by natural selection and biogeographic laws over millennia. While seemingly idiosyncratic, all are surely explainable given more data about the region and attributes of where the Malaise trap samples were run. I would call the within and across regional differences syncretistic not stochastic. The former embraces the idea that each region has a different history and set of forcings.

RESPONSE: We agree with your assessment. Stochastic here refers to the fact that we just don't know the exact mechanism with the provided data, not that the reason is random chance. We have edited the text to better reflect the narrative for the reader and clarify that regional-specific processes are likely driving differences at the regional level (line 215-216).

One factor that I believe is important in determining both alpha diversity and beta diversity that was not mentioned was the paleoclimatic/historical stability of an ecosystem or region. I just returned from Mozambique. I was stunned by the number of flightless beetles—which no doubt = elevated beta diversity. There are almost no flightless beetles in Connecticut, a state that was recently glaciated. This can only happen where a site has been largely stable through antiquity. I recall hearing the same is true for Hemiptera in South Africa, i.e., unparalleled degree of flightlessness there. Stability promotes both alpha and beta diversity. The species diversity and tiny range sizes of plants across the fynbos is a third example. Some of the exceptional differences in regional patterns could relate to antiquity + stability within and across these areas.

RESPONSE: We have included this aspect into the discussion line 303-305

Another major determinant of local/regional alpha and beta diversity is topography, and such could have a major influence of within and between regional patterns. Brazil and Columbia may have the richest biodiversity in the world because they have biotically distinct mountain ranges. Africa has only modest topography by comparison. Topographical differences surely underlie part of the “stochasticity” in the results. Might species/area (island biogeography) account for the non-conforming results from Oceania?

RESPONSE: Thank you, we have expanded the discussion to include regional topological variation as an alternative means to explain the results.

I bring these up as a way of reinforcing my point that the results from different regions likely have sound biogeographic/ecological explanations and that they are not so much stochastic as syncretistic in nature.

RESPONSE: We agree and have removed the term to avoid confusion.

A bit more discussion about what is driving the non-conforming regional patterns would be welcome. Overall the manuscript is Spartan, with just one page of results and < three pages of discussion. Perhaps a bit more discussion about the patterns, drivers, exceptions, and taxonomic implications would be welcome.

RESPONSE: We have included more discussion text on the non-conforming regional patterns. The discussion length has increased to 3.5 pages.

Might some self-assessment of the limitations of the study would be helpful to future workers. Sampling was sparse in South America and Africa. It doesn't appear traps were run in the more diverse (mountainous) regions of either continent. What results were most perplexing and could be better evaluated with more data? I would be interested in getting the author's views of what would be needed to get why some of the regional patterns did not conform to expected patterns.

RESPONSE: We agree and have noted that additional sampling with consideration for increased sampling effort in less studied regions. We also note that given the trends in regional biodiversity observed here that future sampling should look to assess within regional variation, considering regional variation in habitat, landuse, elevation and seasonality. (line 262-267)

Minor matters: alpha and beta diversity are not hyphenated. Hyphen use across the manuscript needs to be considered by someone skilled in the Queen's English (grin)—that would not be me. I noticed several novel and incorrect usages across the introduction.

RESPONSE: Hyphenation is used in compound words or phrases to clarify the relationship between the words and to make the meaning of the term more understandable. In the case of "alpha-diversity," the hyphen helps to indicate that "alpha" and "diversity" are combined to form a single concept related to biodiversity. Additionally, hyphenation can help prevent confusion or misinterpretation of the term. As such, we would prefer to keep the hyphenated usage, but will defer to the editor if there is a need to remove them.

In same vein, when to use between, among, and across at various points in the narrative could be tuned up. Any comparisons involving more than two entities should be using among (or across) rather than between.

RESPONSE: We have checked the document for grammar and edited as suggested

Southern Hemisphere should be capitalized; especially if Equator is capitalized. Both are recommended by Chicago Style Manual.

RESPONSE: Edited, thank you.

Lines 147-149: The estimate of over 2 million is ridiculously low and out of step with modern estimates. At the very least it should be adding the descriptor "metazoans." Or it should say more than 1.5 million described species. This is an arthropod paper and thus is better grounded to Stork (2018). The number is well north of 5.5 million.

RESPONSE: Thank you, we have edited as suggested

Stork, N. E. (2018). How many species of insects and other terrestrial arthropods are there on Earth? *Annual Review of Entomology*, 63, 31-45.

García-Robledo, C. E. K. Kuprewicz, C. S. Baer, E. Clifton, G. G. Hernández, and D. L. Wagner. 2020. The Erwin Equation of Biodiversity: From little steps to quantum leaps in the discovery of tropical diversity. *Biotropica*. 52: 590-597.

Overall, this is a terrific effort of wide interest and can go forward with little modification.

David Wagner

RESPONSE: Thank you so much for your kind assessment

Reviewers' comments:

Reviewer #1 (Remarks to the Author):

Summary:

The authors have made an effort to address all the suggested comments. I have a few minor comments remaining, mainly for clarification or spelling errors etc as I happened to spot them.

The format of the data has been improved for possible future users and the associated code is now much more clearly documented.

There seems to be a mismatch between negative and positive responses described in the text and the direction of the coefficients shown in Supplementary Figure 3. Perhaps I am misinterpreting this but it is rather confusing for a reader trying to match up what is presented in the text and what is shown in accompanying figures.

Introduction:

130: missing "factors" after stochastic. This also looks different to the suggested edit given in the response document. Please ensure that the sentence reads clearly whichever form you choose.

132: Suggest removing "the amazing level of" and just stating that global biodiversity is essential for life.

140: replace "with over 30" with "there are over 30", otherwise the sentence doesn't read well.

195: This new sentence/paragraph could be better integrated. I suggest moving it before the sentence starting "Hence, comparisons" in line 193. Could a reference be added to this sentence as well to highlight some of these key instances?

Results:

207: suggest replacing "diversity" with "range" in the first instance to reduce repetition.

Table 1: Could table one include the direction of the responses as well as the significance? These are described in some places in the text but not in all of them.

Spelling corrections required in supplementary figure legends:

- Supplementary figure 4: The x-axis reads "Sbsolute" rather than "Absolute".
- Supplementary figure 2: Richness difference is described as richness different.
- Supplementary figure 3: negative associated should read negative association.

Line 227 – the sentence still states that a p value of 0.4 is significant.

Discussion:

242: The global malaise trap program is first mentioned in the discussion. This should be introduced earlier either in the last paragraph of the introduction or first paragraph of the results.

244: the description of the main finding here is not consistent with the description in the results section which describes a decrease with absolute latitude rather than an increase with decreasing latitude. Please can the results descriptions be given in a consistent way between the two sections to help with continuity.

252: South American should be South America

256: "to build upon the trends in assess in our finding" doesn't make sense, perhaps just "build upon our findings"

The section on small sample sizes added near the beginning of the discussion feels out of place here, and separates two sections focussed on the main results. Perhaps it would be better located at the end of that first part of the discussion.

Lines 325 – 329: this part reads more like results than discussion, since these have not been presented elsewhere I think they could be moved to the first section of the results.

Data and Code

The data are much easier to understand now, thank you for making those changes.

Thank you also for improving the code organisation and descriptions in the github repo.

Reviewer #2 (Remarks to the Author):

The authors have well adressed all my concerns and comments. The only little detail I found was that I wonder if the last part of the last sentence of the Abstract was deleted on purpose.

Congratulations to this achievement!

Reviewer #3 (Remarks to the Author):

I read through the revised manuscript, paying special attention to areas where new text was proposed, as well as the detailed response to the three reviews. The new text is improved and adds in some of the recommendations that I made for the first draft.

The new edits contain grammatical problems, there is inconsistent use of Oxford comma throughout, and failure to make changes suggested to the first draft (e.g., capitalization of Northern Hemisphere), which collectively suggests a hasty turnaround, with lack of involvement of many of the 75 authors. Were it me I'd run the present version through a few more co-authors for clarity, more consistent grammar, and to excise some of the text redundancy.

Many of my edits and wording suggestions have been added to the PDF, which I am returning.

A paragraph about the disparity of the results within a biogeographic region such as Australia, Oceania, or SE Asia to give readers a tangible understanding as to why trends broke down within a region would be welcome. Most readers would be interested in what geological, ecological, climatic, area-effect, and time-effect processes are in play within a region that confound within-region patterns.

Likewise, a short paragraph about which taxa contributed the most BINs to insect diversity in the Discussion would be of broad interest. I am sure this matter will be treated in other papers, but at least a summation or preview here, with some indication of global patterns (across and within regions), would be appreciated by readers. Most readers and text book tout Coleoptera as the most biodiverse lineage of metazoans.

I am much impressed with study overall, the magnitude of the data, the Herculean efforts behind the data, and the principal findings. I am happy to see the manuscript move forward.

David L. Wagner

Reviewers' comments:

Reviewer #1 (Remarks to the Author):

Summary:

The authors have made an effort to address all the suggested comments. I have a few minor comments remaining, mainly for clarification or spelling errors etc as I happened to spot them.

The format of the data has been improved for possible future users and the associated code is now much more clearly documented.

There seems to be a mismatch between negative and positive responses described in the text and the direction of the coefficients shown in Supplementary Figure 3. Perhaps I am misinterpreting this but it is rather confusing for a reader trying to match up what is presented in the text and what is shown in accompanying figures.

RESPONSE: We have improved the organization of supplementary figure 3 to make the associations easier to match with the main text. The figure legend has been updated as well for clarity and we have included the direction of each coefficient trend in Table 1 as requested.

Introduction:

130: missing "factors" after stochastic. This also looks different to the suggested edit given in the response document. Please ensure that the sentence reads clearly whichever form you choose.

RESPONSE: We have edited this to read "stochastic processes," as suggested by reviewer 3

132: Suggest removing "the amazing level of" and just stating that global biodiversity is essential for life.

RESPONSE: Edited as suggested

140: replace "with over 30" with "there are over 30", otherwise the sentence doesn't read well.

RESPONSE: Edited as suggested

195: This new sentence/paragraph could be better integrated. I suggest moving it before the sentence starting "Hence, comparisons" in line 193. Could a reference be added to this sentence as well to highlight some of these key instances?

RESPONSE: We have repositioned the sentence as suggested and included a reference.

Fine, P. V. (2015). "Ecological and evolutionary drivers of geographic variation in species diversity." Annual Review of Ecology, Evolution, and Systematics 46: 369-392.

Results:

207: suggest replacing "diversity" with "range" in the first instance to reduce repetition.

RESPONSE: edited as suggested.

Table 1: Could table one include the direction of the responses as well as the significance? These are described in some places in the text but not in all of them.

RESPONSE: We have included the direction of the response coefficients as requested. Significance is already indicated by bold face.

Spelling corrections required in supplementary figure legends:

- Supplementary figure 4: The x-axis reads “Sbsolute” rather than “Absolute”.
- Supplementary figure 2: Richness difference is described as richness different.
- Supplementary figure 3: negative associated should read negative association.

RESPONSE: Thank you, each has been adjusted as suggested.

Line 227 – the sentence still states that a p value of 0.4 is significant.

RESPONSE: Corrected to 0.04 as depicted in Table 1

Discussion:

242: The global malaise trap program is first mentioned in the discussion. This should be introduced earlier either in the last paragraph of the introduction or first paragraph of the results.

RESPONSE: We have reorganized the text; parts of the methods description for the global malaise trap program have been moved to the results section (Lines 207-215).

244: the description of the main finding here is not consistent with the description in the results section which describes a decrease with absolute latitude rather than an increase with decreasing latitude. Please can the results descriptions be given in a consistent way between the two sections to help with continuity.

RESPONSE: We have improved the organization and consistency of the results.

252: South American should be South America

RESPONSE: Corrected

256: “to build upon the trends in assess in our finding” doesn’t make sense, perhaps just “build upon our findings”

RESPONSE: Edited as suggested

The section on small sample sizes added near the beginning of the discussion feels out of place here, and separates two sections focussed on the main results. Perhaps it would be better located at the end of that first part of the discussion.

RESPONSE: We have moved this paragraph so it now follows the section on regional trends (now line 339-344) to provide a follow-up with regards to the finer scale observations and their implications.

Lines 325 – 329: this part reads more like results than discussion, since these have not been presented elsewhere I think they could be moved to the first section of the results.

Data and Code

The data are much easier to understand now, thank you for making those changes.

Thank you also for improving the code organisation and descriptions in the github repo.

RESPONSE: Thank you

Reviewer #2 (Remarks to the Author):

The authors have well addressed all my concerns and comments. The only little detail I found was that I wonder if the last part of the last sentence of the Abstract was deleted on purpose.

Congratulations to this achievement!

RESEPNSE: Thank you very much for your time, feedback, and support of our work.

Reviewer #3 (Remarks to the Author):

I read through the revised manuscript, paying special attention to areas where new text was proposed, as well as the detailed response to the three reviews. The new text is improved and adds in some of the recommendations that I made for the first draft.

The new edits contain grammatical problems, there is inconsistent use of Oxford comma throughout, and failure to make changes suggested to the first draft (e.g., capitalization of Northern Hemisphere), which collectively suggests a hasty turnaround, with lack of involvement of many of the 75 authors. Were it me I'd run the present version through a few more co-authors for clarity, more consistent grammar, and to excise some of the text redundancy.

Many of my edits and wording suggestions have been added to the PDF, which I am returning.

Response: We have corrected the grammatical issues as suggested in the attached pdf.

A paragraph about the disparity of the results within a biogeographic region such as Australia, Oceania, or SE Asia to give readers a tangible understanding as to why trends broke down within a region would be welcome. Most readers would be interested in what geological, ecological, climatic, area-effect, and time-effect processes are in play within a region that confound within-region patterns.

RESPONSE: We have included a paragraph to address the general disparity of results within biogeographic regions (lines 303-315).

Likewise, a short paragraph about which taxa contributed the most BINs to insect diversity in the Discussion would be of broad interest. I am sure this matter will be treated in other papers, but at least a summation or preview here, with some indication of global patterns (across and within regions), would be appreciated by readers. Most readers and text book tout Coleoptera as the most biodiverse lineage of metazoans.

RESPONSE: We thank the reviewer for this valuable suggestion. We have added a paragraph to the Discussion that summarizes the taxa that contributed the most BINs to insect diversity in our study (line 262-276). Additionally, we have included a brief discussion of the global patterns of insect diversity within and across regions.

I am much impressed with study overall, the magnitude of the data, the Herculean efforts behind the data, and the principal findings. I am happy to see the manuscript move forward.

RESPONSE: Thank you for your time and valuable feedback

David L. Wagner

Comments from the pdf file provided by reviewer 3

Line 115: relied on data for vertebrates, plants and other less diverse taxa.

RESPONSE: Edited as suggested

Line 122: Not sure in contrast is needed. I would reword to state: Species replacement and richness difference patterns varied markedly with and among major biogeographic regions.

RESPONSE: We have removed the "In contrast," from the beginning of the sentence.

Line 124: takes a comma

RESPONSE: We have included a comma

Line 130: processes

RESPONSE: We have added "processes" to the sentence

Line 133: energy and

RESPONSE: We have included "energy and" in this sentence

Line 141: Change more to some as exceptions have long been known

RESPONSE: We have deleted the "more" to simply state "Recent efforts."

Line 150: Three sentences, here starting at line 150, line 155, and line 157 are saying the same things--some of this redundancy should be eliminated.

RESPONSE: We have revised the four sentences describing species replacement and richness difference to reduce redundancy.

Line 154: If you have competition you need to add differing natural enemy complexes. ANTs in particular come to mind all low-elevation ecosystems.

RESPONSE: We have edited the text to read "competition, natural enemy complexes, or historical disturbances"

Line 173: especially across sites with appreciable seasonality

RESPONSE: Yes, we agree.

Line 201: grammatically need to add, which here serve as

RESPONSE: We have added the requested text.

Line 203: Oxford comma used sometimes but not others.

RESPONSE: The oxford comma has been implemented throughout the manuscript.

Line 203: Oxford comma used inconsistently

RESPONSE: Please see previous comment

Line 205: recovered

RESPONSE: "collected" has been replaced with "recovered"

Line 256: assessed

RESPONSE: "assess" has been replaced with "assessed"

Line 288: add comma

RESPONSE: A comma has been added to the sentence

Line 305: are

RESPONSE: "is" has been replaced by "are"

Line 331: maybe delete "the so-called"

RESPONSE: edited as suggested

Line 333: Northern Hemisphere

RESPONSE: edited as recommended